# Peptidylarginine Deiminase Type 2 Predicts Tumor Progression and Poor Prognosis in Patients with Curatively Resected Biliary Tract Cancer

**DOI:** 10.3390/cancers15164131

**Published:** 2023-08-16

**Authors:** Hon-Yi Lin, Chih-Chia Yu, Chen-Lin Chi, Chang-Kuo Wei, Wen-Yao Yin, Chih-En Tseng, Szu-Chin Li

**Affiliations:** 1Department of Radiation Oncology, Dalin Tzu Chi Hospital, Buddhist Tzu Chi Medical Foundation, Chia-Yi 62247, Taiwan; dm126730@tzuchi.com.tw; 2School of Medicine, Tzu Chi University, Hualian 97004, Taiwan; wck@tzuchi.com.tw (C.-K.W.); wen-yaoyin@tzuchi.com.tw (W.-Y.Y.); p121521@tzuchi.com.tw (C.-E.T.); 3Department of Medical Research, Dalin Tzu Chi Hospital, Buddhist Tzu Chi Medical Foundation, Chia-Yi 62247, Taiwan; dl26558@tzuchi.com.tw; 4Department of Pathology, Chiayi Chang Gung Memorial Hospital, Chia-Yi 61303, Taiwan; b9002038@cgmh.org.tw; 5Department of General Surgery, Dalin Tzu Chi Hospital, Buddhist Tzu Chi Medical Foundation, Chia-Yi 62247, Taiwan; 6Metabolic Surgery and Allied Care Center, Dalin Tzu Chi Hospital, Buddhist Tzu Chi Medical Foundation, Chia-Yi 62247, Taiwan; 7Department of Anatomic Pathology, Dalin Tzu Chi Hospital, Buddhist Tzu Chi Medical Foundation, Chia-Yi 62247, Taiwan; 8Division of Hematology-Oncology, Department of Internal Medicine, Dalin Tzu Chi Hospital, Buddhist Tzu Chi Medical Foundation, Chia-Yi 62247, Taiwan

**Keywords:** biliary tract cancer, peptidylarginine deiminase type 2, survival, prognosis

## Abstract

**Simple Summary:**

Biliary tract cancer (BTC) is a highly aggressive malignancy with a poor prognosis; currently, limited biomarkers are available for early diagnosis and effective disease management. Therefore, it is essential to investigate reliable biomarkers for BTC patients. Peptidylarginine deiminase type 2 (PADI2) is a vital factor for post-translational modification (PTM) that catalyzes arginine to citrulline. It is crucial in several pathophysiological processes, such as autoimmune diseases and cancers. We evaluated the clinical significance of PADI2 expression in pathological stage I–III BTC patients. We observed that patients with high PADI2 protein expression are associated with poorer survival, including progress-free survival (PFS), disease-specific survival (DSS), and overall survival (OS). Our study highlighted, for the first time, a significant correlation between the PADI2 higher expression levels and unfavorable prognosis in BTC, which can be used to predict patients’ survival outcomes.

**Abstract:**

(1) Background: PADI2 is a post-translational modification (PTM) enzyme that catalyzes citrullination, which then triggers autoimmune disease and cancer. This study aimed to evaluate the prognostic value of peptidylarginine deiminase 2 (PADI2) protein expression in biliary tract cancer (BTC) patients. (2) Methods: Using immunohistochemistry, the PADI2 protein expression in BTC tissues was analyzed. The correlations between PADI2 protein expression and clinicopathologic characteristics were analyzed using Chi-square tests. The Kaplan–Meier procedure was used for comparing survival distributions. We used Cox proportional hazards regression for univariate and multivariate analyses. From 2014 to 2020, 30 resected BTC patients were enrolled in this study. (3) Results: Patients with high PADI2 protein expression were associated with shorter progress-free survival (PFS; *p* = 0.041), disease-specific survival (DSS; *p* = 0.025), and overall survival (OS; *p* = 0.017) than patients with low PADI2 protein expression. (4) Conclusions: The results indicated that PADI2 protein expression was an independent poor prognostic factor for BTC patients regarding PFS, DSS, and OS.

## 1. Introduction

Biliary tract cancer (BTC) is a group of malignant tumors that develop in the biliary tract, including cholangiocarcinoma (intrahepatic and extrahepatic), ampullary carcinoma, and gallbladder cancers, demonstrating highly aggressive clinical behavior with poor prognosis [1]. BTC is a heterogeneous disease driven by genetic mutations and transcriptional alterations and affects the immune system within the tumor microenvironment (TME) [2]. Thus, tumor heterogeneity has presented a considerable challenge to BTC patients. Curative surgery remains the mainstay of treatment [3]; however, most patients present with advanced unresectable disease at diagnosis and fared worse outcomes. While several biomarkers have been utilized in clinical practice [3,4], the sensitivity and specificity of the methods used clinically for BTC diagnosis still need improvement, particularly during the early disease stage. Therefore, finding valuable and reliable predictive biomarkers for BTC is necessary. 

Peptidylarginine deiminases (PADs) are a group of enzymes that mediate post-translational modifications of proteins and catalyze the conversion of protein arginine residues to citrulline. Five highly conserved PADs exist in humans, including PAD1–4 and PAD 6. Each isotype has a tissue-specific expression pattern [5,6]. PADI2 is widely expressed in various tissues and has been shown to trigger citrullinated antigens causing symptoms in autoimmune diseases [5,6]. It was previously shown that PADI2 is expressed at high levels in neutrophils [7] and macrophages [8] to accumulate within the joint rapidly, so it is particularly relevant in the pathogenesis of rheumatoid arthritis (RA).

Notably, dysregulation of PADI2 has also been linked to promoting the pathogenesis and progression of cancer [9], including breast [10], prostate [11], and colorectal [12] cancer, and has also been suggested to promote tumor metastasis. It has been reported that PADI2 can serve as a potential biomarker in breast cancer, and its inhibition is an appropriate candidate for therapies addressing early-stage disease [13]. Although evidence has been steadily accumulating to indicate a significant prognostic role for PADI2 levels in patients with cancer, the pathogenic features of PADI2 in cancer are still controversial. For example, it has been reported that high expression of PADI2 was significantly associated with shorter overall survival and progression-free survival in patients with ovarian cancer [14]. Conversely, the downregulation of PADI2 was closely associated with poor prognosis in patients with colorectal cancers (CRC) [15].

Moreover, several recent reports have shown that PAD2-catalyzed citrullination of chemokines, such as CXCL 10 [16] and CXCL8 [17], is associated with impaired recruitment of immune cells. In addition, PAD2 has been reported to mediate the regulation of immunity activity and could potentially affect cross-talk between tumor-associated immune cells [18,19]. Thus, PADI2 represents a multifaceted protein that not only involves tumor progression but also affects tumor immunity.

However, to date, the prognostic role of PADI2 expression in BTC patients has not been investigated. Hence, the present study aimed to investigate the correlation among the PADI2 expression level, the clinical–pathological factors, and the outcomes of BTC patients.

## 2. Materials and Methods

### 2.1. Patient Characteristics

From 2014 to 2020, 106 patients were histopathologically diagnosed as BTC, and 76 patients were excluded from the present study due to having an unresectable disease, being medically unfit for operation, or having surgery with a non-curative intent. Finally, tissue samples and data from 30 curatively resected BTC patients were analyzed (Figure 1). All samples were obtained from patients who underwent radical surgery before chemotherapy or radiotherapy, if any. Histopathological diagnoses were confirmed using standard pathological analysis methodologies. The patient’s characteristics were analyzed according to relevant clinical-pathological factors, such as age, sex, pathologic lymph node (pN), pathologic stage, surgical margin, perineural invasion (PNI), lymphovascular invasion (LVI), chemotherapy, and radiotherapy. The tumors were staged at the time of diagnosis based on the 7th–8th American Joint Committee on Cancer (AJCC) TNM classification [20,21]. All BTC samples were examined retrospectively in an anonymous de-identified manner to fit the ethical standard. 

### 2.2. Cancer Treatment

Surgery plays a vital role in treating BTC patients [22]. Radical surgery was conducted first, including procedures of partial liver resection, the Whipple operation, and lymph node dissection. Individually, the specific surgical approach was determined using various factors, such as the tumor’s location, size, and the patient’s overall performance status. After surgical resection, radiotherapy (RT) with or without chemotherapy was conducted individually for patients with positive/close surgical margins or positive nodal disease [23,24]. Advanced irradiation techniques, i.e., intensity-modulated radiation therapy (IMRT) or volumetric-modulated arc therapy (VMAT), were used for delivering irradiation [25]. Depending on pathological reports, the RT doses ranged from 4500 to 6000 cGy in 25–30 fractions. A boost dose after 4500 cGy was delivered to the clip-oriented high-risk surgical bed using the cone-down boost or simultaneous-integrated inner-escalated boost (SIEB) techniques. Simultaneously, SIEB escalated the radiation dose to the high-risk inner target to enhance tumor control and minimized the dose to the surrounding normal tissues to decrease RT side effects [26,27].

### 2.3. PADI2 Immunohistochemical Evaluation

As described previously [21,22,23], paraffin sections were deparaffinized and rehydrated [28,29,30]. Immunohistochemical (IHC) staining was conducted using a Super Sensitive™ Polymer-HRP IHC detection system (Biogenex, San Ramon, CA, USA). The slides were incubated with PADI2 polyclonal antibody (12110-1-AP, Proteintech, Rosemont, IL, USA) at a dilution of 1:50 at 4 °C overnight, followed by a wash, and then incubated with the peroxidase-conjugated secondary antibody (Abcam, Cambridge, UK), showing cytoplasm and nucleus staining. PADI2 was visualized using 3,3′-diaminobenzidine (DAB), and the slides were counterstained using hematoxylin. Oncological pathologists defined the protein expression scores using the multiplied values of stained intensity (0–2) and percentage of positive cells (0–100%). The PADI2 value was applied as a cutoff value to differentiate high or low expression.

### 2.4. Statistical Analysis

We performed statistical analyses using the SigmaPlot software, version 10.0 (Systat Software Inc., San Jose, CA, USA) and SPSS (version 12.0; SPSS Inc., Chicago, IL, USA), accordingly. We used the chi-square test to evaluate the correlation between PADI2 expression and clinicopathological features. The Kaplan–Meier method was conducted to analyze survival curves. We applied the Cox proportional hazards regression model to estimate univariate and multivariate hazard ratios. All hazard ratios were provided with 95% confidence intervals to delineate effect size. *p* values of <0.05 were defined as statistical significance.

## 3. Results

### 3.1. Relationship between PADI2 Expression and Patients’ Clinicopathological Characteristics

Among the 30 patients evaluated in the present study, the association of PADI2 expression with clinicopathological parameters is shown in Table 1. No significant differences in clinicopathological features such as age, sex, pN, pathologic stage, surgical margin, PNI, LVI, chemotherapy, and radiotherapy were found between the PADI2-low and -high expression groups (*p* > 0.05). 

### 3.2. Relationship of PADI2 Expression on Progress-Free Survival (PFS), Disease-Specific Survival (DSS), and Overall Survival (OS)

We performed IHC to detect the protein expression of PADI2 in 30 BTC tissue samples. We observed that the staining of PADI2 was in the cytoplasm and nuclei of BTC tumor cells (Figure 2). The Kaplan–Meier analysis plotted PFS, DSS, and OS curves based on different PADI2 expression levels. We observed that patients with high PADI2 expression demonstrated a shorter PFS (*p* = 0.041), DSS (*p* = 0.025), and OS (*p* = 0.017) than patients with low PADI2 expression (Figure 3). 

### 3.3. Univariate and Multivariate Analysis of PADI2 Expression on Survival

Univariate and multivariate analyses were conducted to determine the prognostic significance of PADI2 as a predictor of survival in patients with BTC (Table 2 and Table 3). In the univariate analysis, the crude hazard ratios (HRs) for PFS, DSS, and OS were 2.869 (95% confidence interval [CI]: 1.002–8.222, *p* = 0.05), 3.519 (95% CI: 1.101–11.250, *p* = 0.034), and 3.399 (95%CI: 1.183–9.766, *p* = 0.023), respectively. The results revealed that the relative level of PADI2 protein expression was correlated with PFS, DSS, and OS. The other clinicopathological features, such as age, gender, surgical margins, pathologic stage, or LVI, did not meaningfully affect prognosis (*p* > 0.05, Table 2). The multivariate analysis revealed that the adjusted HRs for PFS, DSS, and OS were 5.676 (95%CI: 1.226–26.280, *p* = 0.026), 6.166 (95%CI: 1.274–29.840, *p* = 0.024) and 8.449 (95%CI: 1.843–38.740, *p* = 0.006), respectively. For the BTC, the multivariate analysis confirmed that PADI2 protein expression was statistically significant in predicting patient prognosis (Table 3).

## 4. Discussion

According to the national cancer statistics 2020 of the Taiwan Cancer Registry Annual Report, 2629 persons were newly diagnosed with BTC. An annual increase of 5% for adults aged ≥65 years was noted in 2019. BTC does not usually exhibit symptoms in its early stages. However, over 60% to 70% of symptomatic patients are at an advanced stage of BTC, with a poor prognosis and limited treatment options. Despite improvements in diagnosis and therapy, the recurrence rates during and after treatment remain high, and the average 5-year survival rate after treatment was 10% to 30% [3]. 

Because BTC is a highly heterogeneous and complex disease, tumor cells may undergo several molecular changes and therefore can progress toward more aggressive phenotypes and an overall poor clinical prognosis. Therefore, it is essential to identify critical molecular biomarkers for early diagnosis of BTC. Carbohydrate antigen 19-9 (CA19-9) has been regarded as a crucial diagnostic marker in biliary tract cancer [31]; however, its diagnostic potential is limited due to its restricted sensitivity and specificity. Thus, an appropriate biomarker for early detection and prognosis is urgently needed.

The PAD family of enzymes catalyzes citrullination and involves several physiological processes [32]. Among the several isoforms of the PAD family, PADI2, the focus of the present study, is widely distributed across human organs, and its substrates are diverse, including cell structural proteins, immunomodulating molecules, and histones [6]. One of the essential substrates is vimentin, which is a part of the intermediate filament [6]. Vimentin is increasingly considered a canonical marker of epithelial–mesenchymal transition (EMT). EMT is a mechanism for cancer progression and the metastatic process [33]. Thus, at the molecular level, PADI2 has been implicated in EMT-mediated cancer progression. 

EMT is defined as a biological process in which the epithelial cells exhibit the acquisition of invasive mesenchymal features. This process has been reported to play a vital role in cancer progression in that cancer cells become highly aggressive, resulting in short survival of patients or tumor relapse after therapy [34]. PADI2 has been shown to promote EMT during ovarian cancer progression [14] and skin neoplasms [35]. The involvement of EMT processes in cholangiocarcinoma has been proposed previously [36,37]. Moreover, consistent over-expression of EMT-related features has been shown to correlate with poor prognosis in advanced BTC [38]. 

Clinically, lymphovascular invasion (LVSI) represents a crucial characteristic of EMT [39]. LVSI refers to the spread of cancer cells into the lymphatic and blood vessels, allowing them to metastasize to other distant sites. Therefore, the presence of LVSI is a strong clinical predictor of poor prognosis in many types of cancer, including breast [40,41,42], cervical [43,44,45], and endometrial cancer [46,47,48]. Our data showed a higher rate of LVSI in BTC patients with high expression of PADI2 (66.7% [12/18]) than in patients with low expression (25% [3/12], *p* = 0.06; Table 1). This finding might pave the way for further experimental investigations of PADI2 on EMT in BTC.

Notably, PADI2 has been considered to play an essential role in the metastasis and development of cancer [11,12,13]. It has been reported for the prediction of survival outcomes of cancer patients. Clinical studies have shown that PADI2 overexpression was correlated with cancer progression and poor prognosis, e.g., in ovarian or prostate cancer. However, studies investigating the associations of PADI2 expression with patient survival have shown conflicting results. It has been reported that down-regulation of PADI2 was noted in colon tumors and associated with poor prognosis. In addition, the lower expression of PADI2 was significantly associated with recurrence in patients with hepatocellular cellular carcinoma who underwent surgical resection.

Nevertheless, no research has evaluated the correlation between PADI2 expression and prognosis in BTC patients. It is also unknown whether PADI2 is a significant risk factor causing BTC progression and impacts survival in cancer patients. To the best of our knowledge, the present work is the first study to evaluate the clinical role of PADI2 in BTC. The present study used IHC to demonstrate that PADI2 expression levels are associated with outcomes in BTC patients. The Kaplan–Meier survival analysis showed that patients with high expression of PADI2 predicted poorer PFS, DSS, and OS than patients with low PADI2 expression. These findings implicate that up-regulated PADI2 may promote BTC aggressiveness and relate to poor prognosis. The multivariate analysis indicated that PADI2 positivity was an independent prognostic factor for BTC PFS, DSS, and OS.

The present study harbors several limitations. First, BTC is a relatively rare malignancy. Due to the limited number of included patients, the clinicopathological roles of PADI2 in the recurrent and metastatic risk in BTC patients were not investigated. Second, the retrospective study nature may harbor investigating bias [49] and limit additional bio-target analysis, such as caspases, IL1-beta/IL-18, and chemokines. Hence, prospective studies with larger sample sizes and an enzyme-linked immunosorbent assay (ELISA) or a mass spectrometry analysis are warranted to validate our results. Third, the underlying molecular mechanisms of PADI2 in BTC should be further investigated in vitro or in vivo to elucidate the role of PADI2 in BTC progression. 

## 5. Conclusions

The present study demonstrated that PADI2 proteins were substantially highly expressed in BTC tumor tissues, and PADI2 could be an independent prognostic biomarker for predicting clinical outcomes in BTC patients. Our study highlighted, for the first time, a considerable correlation between the high PADI2 expression on IHC stains and unfavorable prognosis in BTC, which may be clinically applied to predict tumor control and patients’ survival. Moreover, the present study might pave the way for explorations on whether high PADI2 expression plays a role in enhancing LVSI via mediating EMT-related signaling pathways, such as JAK2/STAT3, WNT/β-catenin, and AKT/mTOR, to influence the prognosis of patients with BTC. 

## Figures and Tables

**Figure 1 cancers-15-04131-f001:**
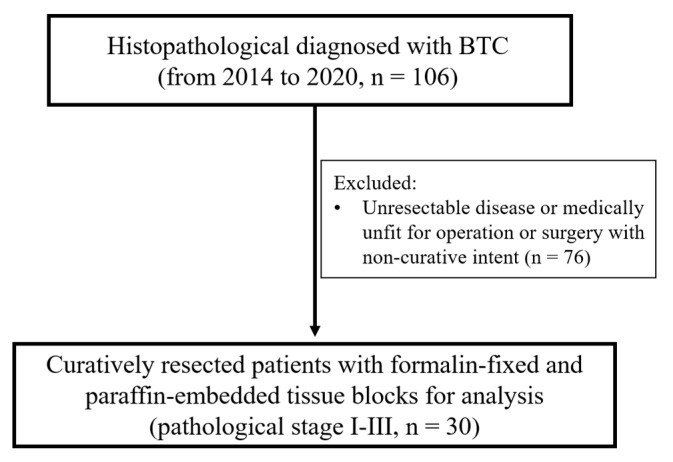
Flowchart of patient allocation.

**Figure 2 cancers-15-04131-f002:**
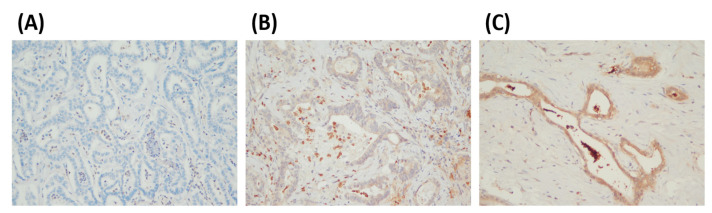
Representative micrographs show BTC’s immunohistochemical (IHC) scores of PADI2 expression (on a scale of 0–2). (**A**) IHC score 0, (**B**) IHC score 1, and (**C**) IHC score 2 (magnification, ×200).

**Figure 3 cancers-15-04131-f003:**
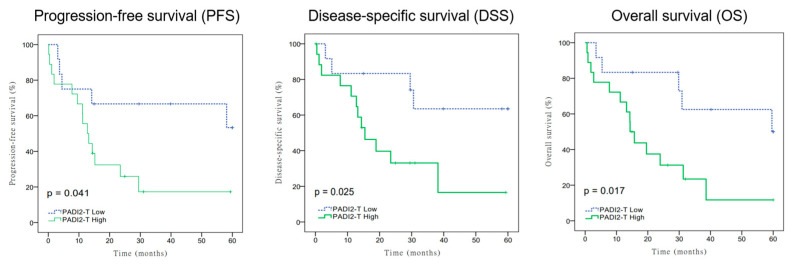
Kaplan–Meier curves show progression-free survival (PFS), disease-specific survival (DSS), and overall survival (OS) based on PADI2 levels.

**Table 1 cancers-15-04131-t001:** Patient characteristics of 30 BTC patients according to the expression level of PADI2.

	PADI2				
Variables	Low Expression (n = 12)		High Expression (n = 18)		*p*-Value
Median age (IQR), years	71.50	(60.5, 76.5)	75.00	(66.5, 77.0)	
Age: n (%)					0.392
<65	4	57.1%	3	42.9%	
≥65	8	34.8%	15	65.2%	
Gender: n (%)					0.710
Male	7	46.7%	8	53.3%	
Female	5	33.3%	10	66.7%	
pN: n (%)					0.465
N0	8	47.1%	9	52.9%	
N1–2	4	30.8%	9	69.2%	
p-Stage: n (%)					0.704
Stage I–II	9	42.9%	12	57.1%	
Stage III	3	33.3%	6	66.7%	
Surgical margins					0.456
<3 mm	5	33.3%	10	66.7%	
≥3 mm	7	46.7%	8	53.3%	
Perineural invasion: n (%)					0.135
Present	4	25.0%	12	75.0%	
Absent	8	57.1%	6	42.9%	
Lymphovascular invasion: n (%)					0.060
Present	3	20.0%	12	80.0%	
Absent	9	60.0%	6	40.0%	
Chemotherapy: n (%)					0.066
No	12	48.0%	13	52.0%	
Yes	0	0.0%	5	100.0%	
Radiotherapy: n (%)					0.130
No	12	46.2%	14	53.8%	
Yes	0	0.0%	4	100.0%	

**Table 2 cancers-15-04131-t002:** Univariate analysis of overall survival (OS), disease-specific survival (DSS), and progression-free survival (PFS).

Variables	Dichotomized Units	OS			DSS			PFS		
		HR	(95% CI)	*p*-Value	HR	(95% CI)	*p*-Value	HR	(95% CI)	*p*-Value
Age	≥65 vs. <65	0.765	(0.274–2.133)	0.609	0.842	(0.270–2.626)	0.767	0.701	(0.252–1.953)	0.497
Gender	Male vs. Female	1.143	(0.463–2.824)	0.772	1.021	(0.379–2.751)	0.968	1.018	(0.413–2.508)	0.969
Surgical margins	≥3 mm vs. <3 mm	0.517	(0.206–1.296)	0.159	0.407	(0.146–1.137)	0.086	0.623	(0.250–1.556)	0.311
p-Stage	III vs. I–II	1.762	(0.690–4.501)	0.236	1.925	(0.695–5.330)	0.208	1.659	(0.650–4.230)	0.290
Lymphovascular invasion	Present vs. Absent	1.826	(0.700–4.763)	0.219	1.824	(0.647–5.142)	0.255	1.515	(0.586–3.920)	0.392
PADI2	High vs. Low	3.399	(1.183–9.766)	0.023	3.519	(1.101–11.250)	0.034	2.869	(1.002–8.222)	0.050

HR, hazard ratio; CI, confidence interval.

**Table 3 cancers-15-04131-t003:** Multivariate analysis of overall survival (OS), disease-specific survival (DSS), and progression-free survival (PFS).

Variables	Dichotomized Units	OS			DSS			PFS		
		HR	(95% CI)	*p*-Value	HR	(95% CI)	*p*-Value	HR	(95% CI)	*p*-Value
Age	≥65 vs. <65	0.476	(0.111–2.033)	0.316	0.470	(0.096–2.300)	0.351	0.490	(0.119–2.013)	0.322
Gender	Male vs. Female	0.932	(0.256–3.392)	0.915	0.827	(0.204–3.355)	0.790	0.850	(0.248–2.917)	0.797
Surgical margins	≥3 mm vs. <3 mm	1.115	(0.232–5.365)	0.892	0.655	(0.129–3.330)	0.610	1.511	(0.306–7.456)	0.613
p-Stage	III vs. I–II	2.841	(0.564–14.321)	0.206	2.201	(0.428–11.334)	0.345	3.066	(0.604–15.556)	0.176
Lymphovascular invasion	Present vs. Absent	0.280	(0.063–1.247)	0.095	0.419	(0.091–1.925)	0.264	0.365	(0.085–1.571)	0.176
PADI2	High vs. Low	8.449	(1.843–38.740)	0.006	6.166	(1.274–29.840)	0.024	5.676	(1.226–26.280)	0.026

HR, hazard ratio; CI, confidence interval.

## Data Availability

All data supporting the conclusions of this article are available upon request.

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
