# Peer review of "Peptidylarginine Deiminase Type 2 Predicts Tumor Progression and Poor Prognosis in Patients with Curatively Resected Biliary Tract Cancer"

_cancers, 2023, doi:10.3390/cancers15164131_

Round 1

Reviewer 1 Report

Dear Sir

 I have read with interest the manuscript entitledPeptidylarginine deiminase type 2 predicts tumor progression and poor prognosis in patients with curatively resected biliary tract cancer” by Hon-Yi Lin and coll.

 In this paper, Authors have investigated by immunochemistry analysis the prognostic value of the PAD12 protein expression in biliary tract cancer (BTC) patients. Data show that an high expression of the PAD12 markers significantly correlated with worse  PFS, DSS and OS suggesting that PAD!” expression is an independent porr prognosis factor in this category of tumors.

The manuscript is correctly written planned a nd discussed and the raised problem is of value; The statistic analysis is of good level.

I have only very few concerns;

1)    In the Back ground  of the abstract they should briefly explain what is the relevance of PAD12.

2)    The discussion, although is very well developed , must be shortened.

Dear Sir

 I have read with interest the manuscript entitledPeptidylarginine deiminase type 2 predicts tumor progression and poor prognosis in patients with curatively resected biliary tract cancer” by Hon-Yi Lin and coll.

 In this paper, Authors have investigated by immunochemistry analysis the prognostic value of the PAD12 protein expression in biliary tract cancer (BTC) patients. Data show that an high expression of the PAD12 markers significantly correlated with worse  PFS, DSS and OS suggesting that PAD!” expression is an independent porr prognosis factor in this category of tumors.

The manuscript is correctly written planned a nd discussed and the raised problem is of value

I have only very few concerns;

1)    In the Back ground  of the abstract they should briefly explain what is the relevance of PAD12.

2)    The discussion, although is very well developed , must be shortened.

I suggest a ligth editing

Author Response

Dear Editors and Reviewers: 

Enclosed, please find the revised manuscript entitled " Peptidylarginine deiminase type 2 predicts tumor progression and poor prognosis in patients with curatively resected biliary tract cancer" that we re-submitted to the Cancers as an Original Research Article.

First, thank you for inviting us to submit a revision of the present work. We replied to all comments on the following pages accordingly. All revised sites in the present manuscript were marked with red color.

All authors have read and approved the re-submitted manuscript. The revised manuscript is significant enough to meet the scope of the Cancers. Thank you for your attention, and we look forward to hearing good news from you soon.

Yours Sincerely,

Szu-Chin Li

Reviewer 2 Report

The author presented an interesting study entitled “Peptidylarginine Deiminase Type 2 Predicts Tumor Progression 2 and Poor Prognosis in Patients with Curatively Resected Biliary  Tract Cancer”. The study is clinically relevant. However, it requires an additional modification to consider in the Cancers Journal.

Major Comment:

1)  Since the study does not use any in-vitro data to support the human data. In addition to IHC data, it requires an additional method for PADI2 detection in the current study.

In addition to Immunohistochemistry (IHC), the Identification of Peptidylarginine Deiminase Type 2 (PADI2) can be detected and studied in various ways to understand its expression, activity, and function. Here are some common methods used for the detection of PADI2:

a) Western blotting: Western blotting is a widely used technique to detect specific proteins. Protein extracts from cells or tissues are separated by gel electrophoresis, transferred to a membrane, and probed with antibodies against PADI2. This method allows for the determination of PADI2 protein levels and can provide information about its molecular weight and potential post-translational modifications.

b)  Enzyme-Linked Immunosorbent Assay (ELISA): ELISA is a sensitive technique used to measure the concentration of a specific protein in a sample. PADI2-specific antibodies are used to capture and detect PADI2 molecules present in the sample. This method can be used to quantify PADI2 levels in biological fluids, such as blood or urine, and can be helpful in biomarker discovery and disease diagnosis.

c)   Real-time Polymerase Chain Reaction (RT-PCR): RT-PCR is a molecular technique used to measure gene expression levels. It can be employed to quantify PADI2 mRNA levels in cells or tissues. RT-PCR provides information about the transcriptional activity of the PADI2 gene and can help determine if its expression is altered under different conditions or in disease states.

d)  Mass spectrometry: Mass spectrometry is a powerful analytical technique used to identify and quantify proteins in complex biological samples. It can be utilized to detect and characterize PADI2 protein and its post-translational modifications, such as citrullination. Mass spectrometry-based approaches provide detailed information about the protein structure and modifications.

The reviewer suggests professional language editing assistance or engaging a native English speaker to review and improve the language.

Author Response

(The authors gave the same response as above.)

Reviewer 3 Report

The manuscript entitled "Peptidylarginine Deiminase Type 2 Predicts Tumor Progression 

and Poor Prognosis in Patients with Curatively Resected Biliary Tract Cancer" by Hon-Yi Lin be accepted for publication with minor comments. 

The author should quantify the Caspases  expression in the cancer patient samples. 

Also, IL1beta, and IL-18 cytokine expression measurement is required. 

The conclusion required more detailed information. 

The PDT2 role in HCC should explained in background/introduction 

The manuscript entitled "Peptidylarginine Deiminase Type 2 Predicts Tumor Progression 

and Poor Prognosis in Patients with Curatively Resected Biliary Tract Cancer" by Hon-Yi Lin be accepted for publication with minor comments. 

The author should quantify the Caspases  expression in the cancer patient samples. 

Also, IL1beta, and IL-18 cytokine expression measurement is required. 

The conclusion required more detailed information. 

The PDT2 role in HCC should explained in background/introduction 

Author Response

(The authors gave the same response as above.)

Round 2

Reviewer 2 Report

The author acknowledges the suggestion for additional validation of the IHC data; however, in response to the review report, it has been deemed out of scope for the current study. Hence the study cannot be accepted in the current format.

Author Response

Dear reviewer, Please find attached.
